# PROSEC: Fortifying Code LLMs with Proactive Security Alignment

Xiangzhe Xu [* 1]  Zian Su [* 1]  Jinyao Guo [1]  Kaiyuan Zhang [1]  Zhenting Wang [2]  Xiangyu Zhang [1]

## Abstract

While recent code-specific large language models (LLMs) have greatly enhanced their code generation capabilities, the safety of these models remains under-explored, posing potential risks as insecure code generated by these models may introduce vulnerabilities into real-world systems. Existing methods collect security-focused datasets from real-world vulnerabilities for instruction tuning in order to mitigate such issues. However, they are largely constrained by the data sparsity of vulnerable code, and have limited applicability in the multi-stage post-training workflows of modern LLMs. In this paper, we propose PROSEC, a novel proactive security alignment approach designed to align code LLMs with secure coding practices. PROSEC systematically exposes the vulnerabilities in a code LLM by synthesizing vulnerability-inducing coding scenarios from Common Weakness Enumerations (CWEs) and generates fixes to vulnerable code snippets, allowing the model to learn secure practices through preference learning objectives. The scenarios synthesized by PROSEC trigger 25× more vulnerable code than a normal instruction-tuning dataset, resulting in a security-focused alignment dataset 7× larger than the previous work. Experiments show that models trained with PROSEC are 25.2% to 35.4% more secure compared to previous work without degrading models' utility.

## 1. Introduction

Large language models (LLMs) capable of generating code based on human instructions have revolutionized programming by significantly facilitating tasks such as code generation (Zhu et al., 2024) and refinement (Zheng et al., 2024;

Guo et al., 2024b). As these models are more widely deployed in productions, their safety becomes increasingly crucial. Insecure code generated by these models has been shown to introduce vulnerabilities, posing risks in real-world applications (Pearce et al., 2021; 2022). Recent studies reveal that even state-of-the-art code LLMs frequently generate insecure code (He & Vechev, 2023; Bhatt et al., 2023; He et al., 2024), highlighting the urgent need for the alignment with secure coding practices.

Enhancing the ability of code LLMs to generate secure code necessitates additional design considerations in their post-training stages, similar to the alignment of general safety, truthfulness, and ethical considerations (Ganguli et al., 2022; Liu et al., 2024c; Ji et al., 2024; Dubey et al., 2024; Hurst et al., 2024). Early efforts, such as SafeCoder (He et al., 2024), seek to address security concerns during the instruction tuning phase by constructing datasets of vulnerable code and corresponding fixes from GitHub commits. The security-focused dataset is then integrated with standard instruction tuning datasets to teach the pre-trained model to generate secure code while preserving utility. However, instruction tuning-based security alignment with real-world data faces two critical challenges:

**Sparsity of Real-World Vulnerability Data.** Vulnerable code snippets in real-world programs and their fixes are often sparse and highly contextual, limiting the effectiveness and generalizability of training secure code LLMs from real-world vulnerabilities. For instance, SafeCoder collects only 465 entries from 145 million git commits. One crucial underlying reason for the sparsity is that human programmers have already avoided most insecure practices before commits so these processes never appear in web data.

**Limited Applicability in Post-Training Pipelines.** Coupling security alignment with the standard instruction tuning phase restricts its utility in modern LLM training workflows. Code LLMs can undergo multi-stage post-training processes based on human/AI feedback for further performance improvements (Lee et al., 2023; Shao et al., 2024; Liu et al., 2024a). Reverting to the initial instruction tuning stage for security alignment necessitates retraining, which is resource-intensive and risks discarding the benefits of prior post-training efforts.

In this paper, we propose PROSEC, a *proactive* security

*Equal contribution  [1]Department of Computer Science, Purdue University, IN, USA [2]Department of Computer Science, Rutgers University, NJ, USA. Correspondence to: Xiangzhe Xu <xu1415@purdue.edu>, Zian Su <su284@purdue.edu>.

*Proceedings of the 42nd International Conference on Machine Learning*, Vancouver, Canada. PMLR 267, 2025. Copyright 2025 by the author(s).

alignment approach to improving the safety of a code LLM that has been post-trained with substantial efforts. It fortifies code LLMs systematically by intentionally triggering and resolving vulnerabilities during post-training. PROSEC exposes the weakness of a code LLM with synthesized coding scenarios. It samples from the target code LLM all normal code, vulnerable code and the corresponding fix under different generation contexts to construct preference data, and aligns the code LLM to secure coding practices with preference learning objectives, minimizing negative effects to its utility.

To address the challenge imposed by the sparsity of vulnerabilities in real-world code repositories, PROSEC leverages the power of prior knowledge from human and synthesized data. The key observation of PROSEC is that the Common Weakness Enumerations (CWEs) (MITRE, 2023), which abstract diverse program vulnerabilities, offer a generalizable foundation for simulating how vulnerabilities manifest across various coding tasks and programming languages. Specifically, PROSEC synthesizes instructions that may expose the weakness of a code LLM by incorporating CWEs into a standard code instruction tuning dataset with a general LLM. Then, these instructions are further leveraged to synthesize data for security alignment training.

To address the second challenge, PROSEC assumes a fully post-trained target model and adopts an additional preference optimization stage for security alignment, without any intervention in previous post-training stages. Given the synthesized vulnerability-inducing and normal instruction datasets, PROSEC constructs preference data for both secure coding practices and utility preservation. Moreover, PROSEC incorporates heuristic and training dynamics-based data selection, leading to a unified high-quality preference dataset. Due to the generality of preference data and the independence of the extra alignment phase, PROSEC can be easily integrated into various post-training pipelines.

Empirically, the instructions synthesized by PROSEC induce 25 times more vulnerable code than a standard instruction tuning dataset. The alignment dataset generated by the proactive approach is 7 times larger than the SafeCoder dataset. We demonstrate the effectiveness of PROSEC on the PurpleLlama (Bhatt et al., 2023) secure coding benchmark. The models trained with the dataset synthesized by PROSEC are 25.2%–35.4% more secure than those trained with the SafeCoder dataset. We further validate that PROSEC does not harm the utility of code LLMs. We conduct thorough ablation studies to justify the design decisions in PROSEC.

**Main Contributions** Our work makes the following key contributions:

- We introduce a novel post-training security alignment process for code LLMs, which systematically ad-

dresses security risks during code generation.

- We develop an automatic pipeline to synthesize and select proactive security alignment data given a code LLM and vulnerability types in a programming language.

- We publish a dataset of synthesized vulnerability-inducing instructions that can effectively expose the weakness of code LLMs. PROSEC and the dataset are available at https://github.com/PurCL/ProSec. The dataset is different from existing datasets that mainly include (vulnerable) code snippets, allowing easy customization to the distribution of target code LLM.

- Through targeted security alignment, we demonstrate that PROSEC improves the ability of code LLMs to generate secure code without compromising their general code generation capabilities, across multiple models, languages, and vulnerability types.

## 2. Background and Problem Formulation

In this section, we introduce the background and how we formulate the security alignment of code LLMs.

**Code LLM** Consider an instruction following code LLM $\pi_\theta(y|x) = \Pi_i \pi_\theta(y_i|y_{<i}, x)$ that takes user instruction $x$ and generates the code response $y$. Notably, $\pi_\theta$ is post-trained for multiple stages with non-trivial efforts after pre-training.

**Security-Related Coding Practice** To ensure safe usage, the code LLM $\pi_\theta$ needs to effectively incorporate the understanding of certain *security-related coding practices* (e.g., sanitizing inputs to prevent command injections). These practices address a range of commonly encountered issues that, if neglected, can render code vulnerable to exploitation. A widely recognized framework for categorizing such issues is the Common Weakness Enumerations (CWE) (MITRE, 2023), which associates each identified weakness with a set of recommended safe coding practices and common pitfalls to avoid. We denote the set of all programming language $l$ and CWE $c$ combinations of interest as $\mathcal{D}_{\text{cwe}} = \{(l^{(i)}, c^{(i)})\}_{i=1}^N$.

Following previous work (Bhatt et al., 2023), we assume that there exists a static analyzer (Bennett et al., 2024; Wu et al., 2024; Mukherjee et al., 2022; Tang et al., 2024a; Wang et al., 2024a; Meta, 2025; Weggli, 2025) as an oracle to detect whether a snippet of code follows secure code patterns. Specifically, the static analyzer takes as input a code snippet, and outputs a list of detected CWEs. An empty output list implies the given code conforms with the secure coding practices of this organization. Formally, we denote

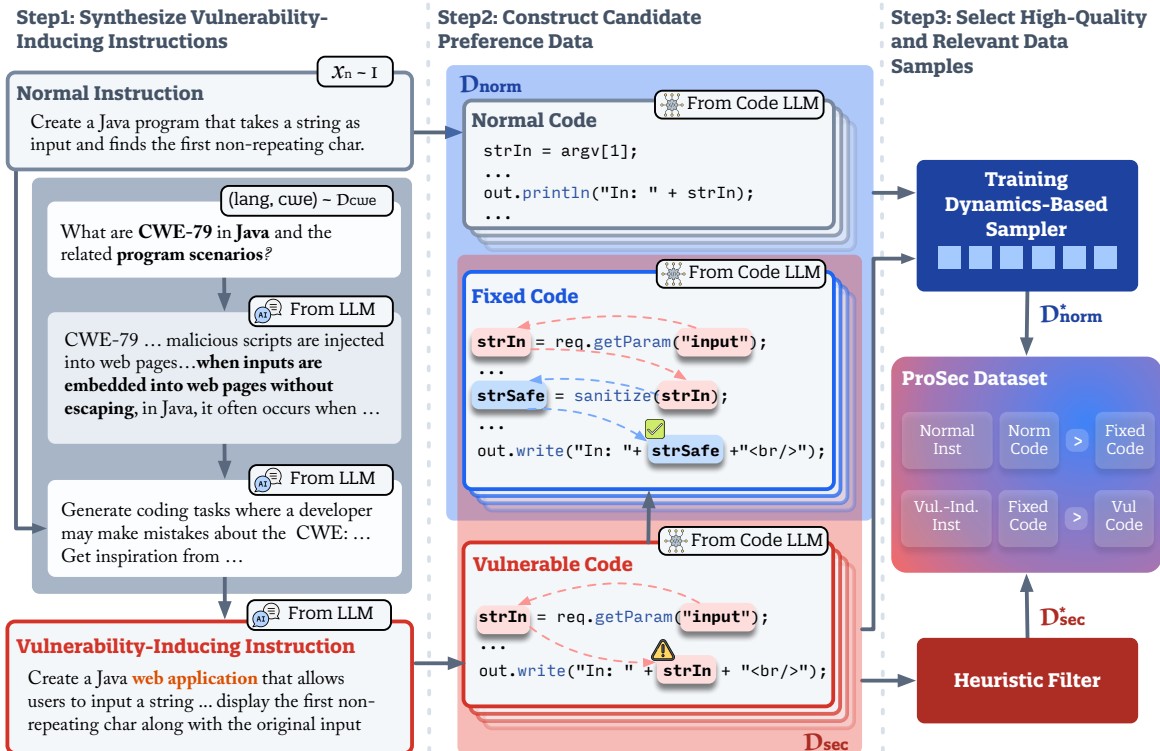

*Figure 1.* PROSEC's data synthesis and selection pipeline. (1) The instruction synthesis stage takes as input a *normal coding instruction* and a ⟨language, CWE⟩ pair, and produces an *vulnerability-inducing instruction* that may trigger the corresponding CWE. (2) The preference data collection stage samples *normal code* and *vulnerable code* snippets from the target model given the normal and vulnerability-inducing instructions respectively. The corresponding *fixed code* snippets are additionally sampled from target model given the vulnerable code and other feedback. The vulnerable instruction, vulnerable code, and fixed code results in $\mathcal{D}_{\text{sec}}$ in the red box. Normal instruction, normal code, and fixed code results in $\mathcal{D}_{\text{norm}}$ in the blue box. (3) The data selection stage leverages a heuristic filter and a training dynamics-based sampler to improve the quality of data in $\mathcal{D}_{\text{sec}}$ and $\mathcal{D}_{\text{norm}}$ respectively and produce the final preference dataset.

the static analyzer as follows with $\mathcal{Y}$ denoting code.

$$S : \mathcal{Y} \rightarrow \emptyset \cup \mathcal{D}_{\text{cwe}} \cup \mathcal{D}_{\text{cwe}}^2 \cup \cdots \mathcal{D}_{\text{cwe}}^N \qquad (1)$$

**Security Alignment of Code LLM** The goal of security alignment in code LLMs is to reduce the likelihood of generating insecure code while preserving its ability to generate functional code that follows user instructions. We consider the security alignment of code LLM as *an additional offline preference optimization stage* conducted after the main training process. This stage leverages a preference optimization objective under the Bradley-Terry (BT) model (Bradley & Terry, 1952; Rafailov et al., 2024): given the dataset $\mathcal{D}_p$, the optimization process minimizes a preference loss function $\mathcal{L}_\theta : \mathcal{X} \times \mathcal{Y}_w \times \mathcal{Y}_l \rightarrow \mathbb{R}$,

$$\theta^* = \underset{\theta}{\arg\min} \sum_{(x, y_w, y_l) \in \mathcal{D}_p} \mathcal{L}_\theta(x, y_w, y_l), \qquad (2)$$

where $x$, $y_w$, and $y_l$ denote a prompt, a preferred/win response, and a less preferred/lose response. Such formula-

tion enables seamless integration with many existing post-training pipelines and avoids retraining.

## 3. PROSEC: Proactive Security Alignment of Code LLMs

In this section, we introduce PROSEC. At a high level, PROSEC is a systematic way of synthesizing and selecting data for the preference optimization of code LLM to guarantee secure code generation while preserving utility. An overview of PROSEC's data synthesis and selection pipeline is shown in Figure 1. We discuss how PROSEC synthesizes vulnerability-inducing instructions in Section 3.1, how it constructs candidate preference datasets in Section 3.2, and how to control the quality of the final alignment dataset via specialized data selection in Section 3.3.

### 3.1. Vulnerability-Inducing Instruction Synthesis

PROSEC's data synthesis begins with a high-quality vulnerability-inducing instruction dataset, intended for later

**Algorithm 1** Vulnerability-inducing instruction generation

---

**input** $\mathcal{D}_{\text{cwe}}$: a set of CWEs, $\mathcal{I}$: a standard instruction dataset

**output** $\mathcal{V}$: a set of vulnerability-inducing instructions. Each entry contains $l, c, x_n, x_v$, denoting the programming language, the CWE, the normal instruction, and the vulnerability-inducing instruction, respectively.

1: $\mathcal{V} \leftarrow \emptyset$
2: **for** $l, c \in \mathcal{D}_{\text{cwe}}$ **do**
3:    $scenario \leftarrow$ query_cwe_definition$(l, c)$
4:    $\mathcal{I}_r \leftarrow$ relevant_instruction$(\mathcal{I}, l, c)$
5:    $\mathcal{V}_0 \leftarrow \emptyset$
6:    **for** $x_n \in \mathcal{I}_r$ **do**
7:       $x_v \leftarrow$ compose$(x_n, scenario, l, c)$
8:       $\mathcal{V}_0 \leftarrow \mathcal{E}_0 \cup \{(l, c, x_n, x_v)\}$
9:    **end for**
10:   $\mathcal{V} \leftarrow \mathcal{V} \cup$ cluster$(\mathcal{V}_0, K)$
11: **end for**

---

sampling of code responses. Existing large-scale coding instruction datasets for standard programming tasks (Wei et al., 2023; BAAI, 2024) are insufficient for this purpose, as many CWEs arise from highly specific coding scenarios underrepresented in these datasets. For instance, CWE-79, illustrated in Figure 1, refers to *Cross-Site Scripting*, where user inputs are embedded into web pages without proper sanitization, allowing attackers to execute arbitrary code in a victim's browser. To reveal a code LLM's limitations in addressing CWE-79, tasks must involve writing web applications. Empirical evidence (Figure 3) shows that only about 0.7% of a standard instruction-tuning dataset can trigger CWEs.

PROSEC address the problem by incorporating prior knowledge of secure coding practices, the CWEs, into the instruction synthesis process. We describe how PROSEC synthesizes vulnerability-inducing instructions in Algorithm 1. Given a programming language and a CWE, PROSEC queries a general knowledge-intensive LLM to enumerate program scenarios that might trigger the CWE in the corresponding language (line 3). In addition, PROSEC selects the normal instructions that are relevant to the programming language from the instruction-tuning dataset (line 4). For each relevant normal instruction, PROSEC then instructs a general LLM to compose the vulnerability-inducing instructions by combining the normal instruction with the program scenarios that may trigger the vulnerability (line 7). The red block in the left part of Figure 1 shows a concrete example. The prompts used are in Appendix B.

We noticed the lack of diversity in LLM generated coding scenarios in our preliminary experiments. Hence, we sample multiple answers for each query with a high temperature, and cluster all instructions relevant to a language and CWE

to $K$ clusters. Only the centroid of each cluster is included in the final instruction dataset, as denoted by line 10 in Algorithm 1. Figure 4 empirically shows that the distribution of the instructions is more diversified after clustering.

## 3.2. Candidate Preference Dataset Construction

Given the synthesized vulnerability-inducing instruction dataset $\mathcal{V}$ and the original instruction dataset $\mathcal{I}$, PROSEC samples two types of candidate preference data from the target model $\pi_\theta$: $\mathcal{D}_{\text{sec}}$ intended to increase the model's ability to generate secure code, and $\mathcal{D}_{\text{norm}}$ to preserve the model's utility.

**Secure Practice Preference Data** Each data sample in the secure coding practice preference dataset $\mathcal{D}_{\text{sec}} = \{(x_v^{(i)}, y_f^{(i)}, y_v^{(i)})\}_{i=1}^{M_s}$ consists of $x_v$, denoting a vulnerability-inducing instruction, $y_v$, the vulnerable implementation, and $y_f$, the counterpart of $y_v$ but with the vulnerability fixed.

PROSEC samples both $y_v$ and $y_f$ from the target model to minimize the negative effects on the model's original distribution during alignment. An important observation in our experiment is that a post-trained model is able to fix an insecure code snippet, given the vulnerabilities identified in the insecure code, even though it makes mistakes with only the vulnerability-inducing instruction. Based on the observation, PROSEC first collects vulnerable code snippets by sampling the target model's response on the vulnerability-inducing instructions. Then it asks the target model to fix the identified vulnerabilities.

Specifically, given a vulnerability-inducing instruction $x_v$, PROSEC samples multiple responses from the target model $\pi_\theta$. Then, it uses the static analyzer $\mathcal{S}$ to check potential insecure coding practices from these responses. For each identified insecure code snippet $y_v \in \{y|S(y) \neq \emptyset \wedge y \sim \pi_\theta(y|x_v)\}$, PROSEC queries the target model with both the code and the knowledge (language, CWE, and identified issue) about the identified vulnerability, instructing the target model to fix the code. Similarly, PROSEC samples multiple responses from the target model and uses the static analyzer to select the secure fixed ones $y_f \in \{y|S(y) = \emptyset \wedge y \sim \pi_\theta(y|x_v, y_v, l, c)\}$. The final $\mathcal{D}_{\text{sec}}$ includes multiple paired $y_f$ and $y_v$s, which will be selected later, for each instruction $x_v$.

Note that an alternative design is to use the static analyzer to identify *both* $y_v$ and $y_f$ from the responses to a vulnerability-inducing instruction, instead of fixing $y_v$ to get $y_f$. We show an example in Appendix D.2.

**Utility Preservation Preference Data** Empirically, we find that the aligned model may undesirably overemphasize features that only appear in the win samples $x_f$ of

$(x_v, y_f, y_v) \in \mathcal{D}_{\text{sec}}$ when only trained on secure practice preference data. For example, suppose that the API `sanitize` in *Fixed Code* of Figure 1 only appears in fixed code snippets (i.e., $y_f$). A model trained exclusively with $\mathcal{D}_{\text{sec}}$ may overemphasize this API, incorporating it in all implementations regardless of the programming context. That is undesirable because the sanitation would cause unexpected behavior for normal coding tasks that print strings to the command line.

To mitigate the problem, we propose to create a companion dataset $\mathcal{D}_{\text{norm}} = \{(x_n^{(i)}, y_n^{(i)}, y_f^{(i)})\}_{i=1}^{M_n}$ for $\mathcal{D}_{\text{sec}}$. $\mathcal{D}_{\text{norm}}$ consists of normal instructions $x_n$, win responses $y_n$ and lose responses $y_f$. $x_n$ is the normal instruction corresponding to the vulnerability-inducing instruction $x_v$ and its response $y_v$. Such preference data strengthens normal response under normal instructions, while suppressing the likelihood of $y_f$ being generated in normal scenarios, thus preserving utility. Similar to $\mathcal{D}_{\text{sec}}$, we also collect multiple response pairs for each instruction in $\mathcal{D}_{\text{norm}}$ for later selection which we will discuss next.

### 3.3. Preference Data Quality Control

We propose a heuristic-based data selection process for $\mathcal{D}_{\text{sec}}$ and a training dynamics-based one for $\mathcal{D}_{\text{norm}}$ to control the quality of the final preference data for alignment training.

**$\mathcal{D}_{\text{sec}}^*$ Selection with Heuristics** For code responses in $\mathcal{D}_{\text{sec}}$, we first use AST parsers to perform a light-weight check on code syntax. We discard code snippets that have syntax errors. As discussed in Section 3.2, we use the static analyzer to ensure the fixed code snippet does not contain vulnerabilities. Moreover, we find the target model may skip unchanged code blocks when generating fixed code snippets. We use keywords (e.g., "remain unchanged") and a minimal length threshold to filter out that noise. Finally, we increase the data diversity by de-duplicating data entries with similar fixed code. We use fuzzy ratio [1](based on Levenshtein-distance) to measure similarity.

**$\mathcal{D}_{\text{norm}}^*$ Selection with Training Dynamics** PROSEC captures the influence of each $(x_n, y_n, y_f) \in \mathcal{D}_{\text{norm}}$ by computing the correlation between two measures w.r.t. training dynamics. Specifically, we first obtain a series of checkpoints $\{\theta_1, \cdots, \theta_T\}$ by performing preference optimization on $\pi_\theta$ with the full $\mathcal{D}_{\text{sec}}$ as a warm-up dataset. Then we compute the following $\boldsymbol{m}_f$ and $\boldsymbol{m}_g$ which are defined as two sorts of training dynamics in our scenario,

$$\boldsymbol{m}_f = [f(\cdots, \theta_1), \cdots, f(\cdots, \theta_T)] \quad (3)$$
$$\boldsymbol{m}_g = [g(\cdots, \theta_1), \cdots, g(\cdots, \theta_T)] \quad (4)$$

Here, $f$ and $g$ are defined as,

$$f(x_n, y_n, y_f, \theta) = r(x_n, y_n, \theta) \quad (5)$$
$$g(x_n, x_v, y_f, \theta) = - r(x_v, y_f, \theta) \quad (6)$$

where $r(x, y, \theta) = \frac{1}{|y|} \log \pi_\theta(y|x)$, and $(x_v, y_f, y_v) \in \mathcal{D}_{\text{sec}}$ is the corresponding secure practice data. The influence score of each $(x_n, y_n, y_f)$ w.r.t. $(x_v, y_f, y_v)$ is therefore

$$\text{Inf}(x_n, y_n, y_f, x_v, y_v) = \text{corr}(\boldsymbol{m}_f, \boldsymbol{m}_g) \quad (7)$$

where $\text{corr}(\cdot, \cdot)$ is the Kendall Tau correlation (Kendall, 1938), which is rank-based and relatively more robust.

We use $\text{Inf}(\cdot)$ to select top-ranking candidate $(x_n, y_n, y_f)$ given $(x_i, y_f, y_v)$ to obtain $\mathcal{D}_{\text{norm}}^*$. The intuition behind this data selection paradigm is that *the most influenced is the most influential* for utility preservation. The dynamic $r(x_n, y_n, \theta)$ in $f$ denotes how the model perceives normal instructions and responses across $\{\theta_1, \cdots, \theta_T\}$, and $r(x_v, y_f, \theta)$ in $g$ denotes how the model becomes more aligned to secure coding practices. As the checkpoints are obtained by the warm-up training with $\mathcal{D}_{\text{sec}}$, a strong correlation between $f$ and $g$, e.g. $r(x_n, y_n, \theta)$ is decreasing while $r(x_v, y_f, \theta)$ is increasing, potentially indicates that target model's ability to generate $y_n$ given $x_n$ is influenced by learning to generate $y_f$ given $x_v$, in other words $y_n$ and $y_f$ are quite relevant conditioned on $x_n$. Therefore, we need to add such utility preservation data in the final dataset to surgically prevent overfitting. Empirically, we find this strategy quite effective in achieving both security and utility.

**Final Preference Dataset** The final dataset for preference optimization is the shuffled mixture of $\mathcal{D}_{\text{sec}}^*$ and $\mathcal{D}_{\text{norm}}^*$.

## 4. Experiment Setup

**Seed Instruction-Tuning Dataset** We use the code-related part of Infinity-Instruct [2] (BAAI, 2024) as our seed instruction dataset for data synthesis.

**Static Code Analyzer** We adopt the static analyzer commonly used by previous work (Bhatt et al., 2023; Liu et al., 2024b) to detect insecure coding practices.

**Test Dataset** We use PurpleLlama (Bhatt et al., 2023) as the test dataset for code model safety. PurpleLlama provides a set of instructions that may trigger errors from a code LLM. We select 38 ⟨language, CWE⟩s from PurpleLlama that are overlapped with SafeCoder, corresponding to 694 test cases. We use the multi-lingual version of Humaneval (Chen et al., 2021; Guo et al., 2024a) and the multi-lingual version of MBPP (Austin et al., 2021) (denoted as MXEval (Athiwaratkun et al., 2022)) as the test dataset for utility.

---

[1]https://pypi.org/project/fuzzywuzzy/

[2]https://huggingface.co/datasets/BAAI/Infinity-Instruct

**Metrics** Following the setup of PurpleLlama, we generate multiple samples for each test instruction, and calculate the ratio of secure code among all generated code samples. We use pass@1 (Chen et al., 2021) as the metric for utility.

**Models and Baselines** We use `claude-3.5-haiku` as the general LLM in our data synthesis pipeline. We synthesize 10k instructions for each CWE and select the most diverse 2k instructions via clustering. The cost to synthesize instructions for each CWE is around 5 USD. We use two well post-trained target models, Phi3-mini-Inst (Abdin et al., 2024) and CodeLlama-7B-Inst (Rozière et al., 2024) in our evaluation. We compare PROSEC with previous SOTA Safecoder from two perspectives. First, SafeCoder is a security-aware instruction-tuning technique. We therefore compare the CodeLlama-7B instruction-tuned by SafeCoder with the CodeLlama-7B aligned from CodeLlama-7B-Inst with the dataset synthesized by PROSEC. Second, Safe-Coder comes with a dataset constructed from real-world vulnerability and fixes. We compare the effectiveness of the SafeCoder dataset with PROSEC synthesized dataset by using both datasets at the alignment stage.

**Optimization** If not otherwise specified, we use SimPO (Meng et al., 2024) as the default preference optimization objective in PROSEC to optimize the model,

$$\mathcal{L}(\theta) = -\mathbb{E}_{(x,y_w,y_l)\sim\mathcal{D}_{\text{norm}}^* \cup \mathcal{D}_{\text{sec}}^*} \quad (8)$$
$$\left[\log\sigma\left(\frac{\beta}{|y_w|}\log\pi_\theta(y_w|x) - \frac{\beta}{|y_l|}\log\pi_\theta(y_l|x) - \gamma\right)\right]$$

where $\beta$ and $\gamma$ are hyperparameters. We also include experiments with other objectives in Appendix D.3 to show the generalizability of the data.

**Warm-up Training for Influence Score** We train each target model on $\mathcal{D}_{sec}$ for 1k steps and leverage checkpoints of every 100 steps to compute the training dynamics for $\mathcal{D}_{\text{norm}}$ data influence score computation.

## 5. Results

We report the main results with PROSEC sampling the top-2 influential among all candidate utility preservation preference data $(x_n, y_n, y_f)$ for each corresponding secure practice data $(x_v.y_f, y_v)$ and further discarding the universally least 20% influential ones within the remaining data. The setting is the same for both Phi3-mini-Inst and CodeLlama-7B-Inst's main experiments.

Our main results with regard to secure code generation and utility are shown in Table 1.

**Secure Code Generation** We can see that for both Phi3-mini-Inst and CodeLlama-7B-Inst, models aligned with the

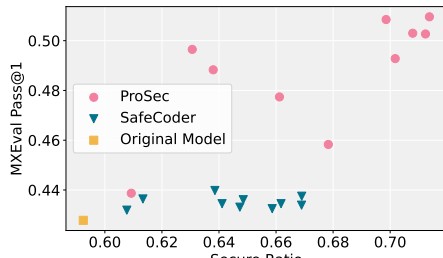

*Figure 2.* How safety and utility of code LLMs change while aligned with different datasets.

PROSEC dataset achieve the most secure results. Specifically, the models aligned with PROSEC are more secure than ones aligned with SafeCoder by 25.2% (33.47 v.s. 44.72) and 35.4% (26.04 v.s. 40.33). That demonstrates PROSEC effectively synthesizes higher-quality data for secure code alignment.

Moreover, for models aligned from CodeLlama-7B-Inst, we can observe that the model aligned with the PROSEC dataset achieves better performance (26.04 v.s. 42.88) than the SafeCoder-Inst model that uses the SafeCoder dataset at the instruction-tuning stage. It demonstrates that enforcing secure coding practices to a post-trained model at the alignment stage is more effective than incorporating them at the instruction tuning stage.

**Effects on Model Utility** For both Phi3-mini-Inst and CodeLlama-7B-Inst models aligned with PROSEC, we can see that their utility performance has no significant downgrades. By contrast, their performance is slightly better than the original model. The improvements on utility might come from the higher complexity of security-related programming scenarios than the ones in a typical instruction-tuning dataset, facilitating models' performance on more challenging tasks. Moreover, we can see that for most cases, models aligned with PROSEC have better utility performance than the models aligned with the SafeCoder dataset.

We further study the effects of alignments on both PROSEC and SafeCoder dataset by visualizing the training trajectories of both alignment training processes. The results are in Figure 2. Specifically, we collect 10 checkpoints for the Phi3-mini-Inst models aligned with the PROSEC dataset and the SafeCoder dataset, respectively. The utility performance is measured by the pass@1 on the MXEval dataset, and the safety is measured by the ratio of secure code generations on the PurpleLlama dataset. Due to resource limitations, we randomly sample subsets of both the MXEval and the PurpleLlama datasets. We can see that for most checkpoints, models trained with PROSEC are consistently more secure than ones trained with SafeCoder. Meanwhile, PROSEC

*Table 1.* Evaluation results for secure code generation and multilingual utility. First three rows denote models aligned from Phi3-mini-Inst and the following three rows denote models aligned from CodeLlama-7B-Inst. PROSEC denotes the alignment dataset is synthesized by PROSEC while SafeCoder denotes the dataset is the SafeCoder dataset. The last row denotes the CodeLlama-7B instruction-tuned with the SafeCoder dataset.

| Model | Vulnerable Code Ratio (%, ↓) | | | | | | HumanEval-Multi (%, ↑) | | | | | MXEval (%, ↑) | | | | |
|---|---|---|---|---|---|---|---|---|---|---|---|---|---|---|---|---|
| | C | C++ | Java | JS | PY | Avg. | C/C++ | Java | JS | PY | Avg. | C/C++ | Java | JS | PY | Avg. |
| Phi3m-Inst | 72.17 | 30.26 | 63.56 | 52.24 | 34.63 | 50.57 | 27.30 | 19.67 | 31.38 | 51.22 | 32.39 | 37.35 | 41.20 | 37.77 | 45.79 | 40.53 |
| w/ SafeCoder | 66.46 | 22.95 | 59.76 | 47.74 | 26.69 | 44.72 | 24.55 | 18.84 | 23.43 | 48.91 | 28.93 | 40.13 | 40.80 | 38.66 | 47.55 | 41.79 |
| w/ PROSEC | 44.27 | 20.74 | 49.09 | 28.21 | 25.05 | **33.47** | 29.74 | 18.39 | 32.36 | 56.11 | **34.15** | 39.00 | 39.26 | 50.10 | 47.75 | **44.03** |
| CLM-7B-Inst | 67.21 | 43.57 | 63.46 | 51.12 | 34.83 | 52.04 | 20.15 | 25.75 | 24.32 | 29.30 | 24.88 | 33.92 | 37.98 | 38.77 | 27.39 | 34.52 |
| w/ SafeCoder | 56.92 | 28.98 | 54.73 | 41.31 | 19.73 | 40.33 | 24.22 | 29.68 | 25.63 | 31.52 | 27.76 | 35.50 | 38.28 | 38.96 | 28.59 | 35.33 |
| w/ PROSEC | 32.50 | 16.67 | 26.94 | 34.10 | 20.00 | **26.04** | 23.96 | 29.35 | 31.49 | 31.27 | **29.02** | 37.43 | 40.33 | 44.04 | 37.78 | **39.89** |
| SafeCoder-Inst | 63.96 | 29.64 | 48.93 | 47.74 | 24.14 | 42.88 | 19.83 | 10.62 | 21.74 | 26.80 | 19.75 | 28.30 | 31.51 | 33.75 | 32.20 | 31.44 |

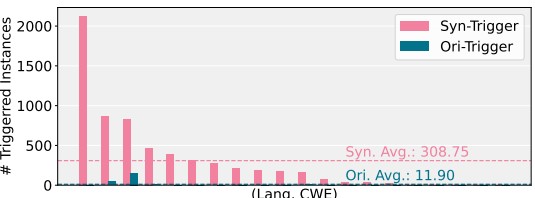

*Figure 3.* Synthesized instructions induce more CWE instances. Each bar denotes the number of vulnerable code instances that trigger the detector for a given language/CWE. We can see that the synthesized instructions induce significantly more vulnerable code instances from the code LLM.

models achieve better utility performance than the Safe-Coder models. That demonstrates PROSEC dataset is more effective than the SafeCoder dataset.

In all, both PROSEC and SafeCoder have limited effects on model utility, while PROSEC is more effective on the model safety.

# 6. Analysis

In this section, we study the design decisions in PROSEC. Due to resource limitations, the evaluation for safety is on a randomly sampled subset of the PurpleLlama dataset.

**Generalizability to Different Models** We evaluate the generalizability of PROSEC by applying it to align three additional models. The results show that PROSEC can consistently improve the security of generated code without harming the utility of an aligned model. Details are in Section D.1 of the appendix.

**Ablation on Vulnerability-Inducing Instruction Synthesis** We illustrate the effectiveness of vulnerability-inducing instructions by showing that they introduce more vulnerable code instances than the original instructions. The

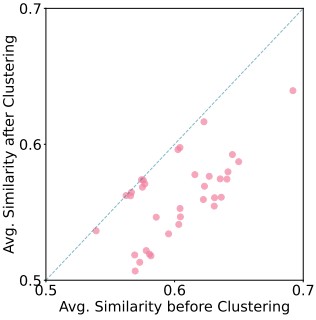

*Figure 4.* Effectiveness of instruction clustering. Each data point denotes a set of synthesized coding instructions for a language/CWE. A larger average similarity indicates lower diversity. We can see that the instructions after clustering are significantly more diversified (i.e., have lower average similarity).

results are visualized in Figure 3, demonstrating that the synthesized instructions indeed induce more vulnerable code snippets.

**Ablation on Instruction Clustering** We study the effectiveness of the instruction clustering by measuring the average similarity between all coding instructions for both the instructions before and after the clustering. The results are shown in Figure 4. We can see that the instruction clustering process indeed makes the synthesized data more diverse.

**Ablation on $\mathcal{D}_{norm}$ and Its Sampling** We compare the proposed $\mathcal{D}_{norm}$ data selection approach with random sampling. Specifically, we fix the $\mathcal{D}_{sec}^*$ in the final preference dataset and sample the same ratio of $\mathcal{D}_{norm}$ for comparison. As shown in Table 2, we can see that with a low ratio of 0.1, the utility of the target model drops significantly, indicating the significance of $\mathcal{D}_{norm}$ to utility preservation. For different ratios of $\mathcal{D}_{norm}$, we can see that PROSEC's sampling leads to more secure models compared to random

*Table 2.* Effectiveness of the data selection algorithm and $\mathcal{D}_{\text{norm}}$. *Random* and PROSEC denote the random selection strategy and the data selection algorithm used in PROSEC, respectively. The second column denotes the ratio of sampled examples from $\mathcal{D}_{\text{norm}}$.

| Strategy | $\mathcal{D}_{\text{norm}}$ ratio | Vul(%, ↓) | Util(%, ↑) |
|---|---|---|---|
| Random | 0.1 | 6.02 | 12.30 |
| PROSEC | 0.1 | 5.92 | 15.28 |
| Random | 0.3 | 32.78 | 41.84 |
| PROSEC | 0.3 | 27.54 | 42.13 |
| Random | 0.7 | 30.92 | 47.26 |
| PROSEC | 0.7 | 25.58 | 45.12 |

*Table 3.* Ablation on which measure to be used for training dynamics-based data influence computation. The $\theta$ in $r(x, y, \theta)$ is omitted here.

| $f$ | $g$ | Vul (%,↓) | Util (%,↑) |
|---|---|---|---|
| $r(x_n, y_n)$ | $-r(x_v, y_f)$ | **23.02** | 45.94 |
| $r(x_n, y_n) - r(x_n, y_f)$ | $-r(x_v, y_f)$ | 23.17 | 45.94 |
| $r(x_n, y_n) - r(x_n, y_f)$ | DECREASE | 27.12 | **46.44** |

sampling, with comparable utility performance. Moreover, observe that the effectiveness of the data selection approach is more prominent when fewer normal data samples are selected (i.e., lower sample ratios), demonstrating its capability in identifying important data samples.

**Ablation on Training Dynamics Options** We also compare different options of training dynamics for the influence score computation. As shown in Table 3, we ablate on both $f$ and $g$. For $f$, as a target model with good utility should also NOT prefer response $y_f$ given input $x_n$, so the $-r(x_n, y_f)$ term can potentially be added to $r(x_n, y_n)$ as an alternate $f$. Results show that this alternative has similar performance to the default one. For $g$, we experiment with a "DECREASE" alternative. Under the context of rank correlation (Kendall, 1938), we denote "DECREASE" as any sequence of monotonic decreasing values. Such correlation as influence score only captures the degrading of utility preservation data but not security. Therefore, we can see that this $g$ leads to the best utility but the worst security in the table. As the major concern in our scenario is the security of the target model, we choose $f = r(x_n, y_n), g = -r(x_v, y_f)$ as our final measure.

**Comparison to Iterative Refinement** An alternative approach to generating secure code is iteratively prompting a code LLM to revise the generated code. The results in Table 4 show that a coding request requires five attempts of fixes to achieve comparable performance with PROSEC, indicating that an agentic workflow incurs higher computational costs and increased latency because it runs a static analyzer for every coding request and may require multi-

*Table 4.* Comparison to iterative refinement. We implement an iterative refinement baseline that employs a static analyzer to verify the security of generated code and instructs the code LLM to revise any insecure code. We use Phi3-mini-Inst as the code LLM. *Max Iterations* denotes the maximum number of revision attempts the system performs to address insecure code.

| | Max Iterations | | | PROSEC |
|---|---|---|---|---|
| | 3 | 5 | 10 | |
| Vul (%,↓) | 31.4 | 26.3 | 19.4 | 25.0 |

*Table 5.* Ablation on the effects of $\mathcal{D}_{\text{norm}}$ and the size of dataset. *Original* denotes the performance of the subject model before security alignment. *SafeCoder* and PROSEC denote the model aligned with SafeCoder and PROSEC dataset, respectively. $+\mathcal{D}_{norm}$ denotes the model aligned with a combined dataset of SafeCoder examples and $\mathcal{D}_{\text{norm}}$ examples drawn from PROSEC. *Subset* denotes the model aligned using a randomly selected subset of PROSEC that matches the SafeCoder dataset in size.

| Setup | Vul (%,↓) | Util (%,↑) |
|---|---|---|
| Original | 40.8 | 42.8 |
| SafeCoder | 33.1 | 43.4 |
| $+\mathcal{D}_{norm}$ | 34.4 | 46.1 |
| PROSEC | 25.0 | 45.2 |
| Subset | 28.9 | 47.0 |

ple queries to the code language model. This design could degrade the user experience in scenarios like code copilots, where swift completions are expected.

**Ablation on the Effects of $\mathcal{D}_{\text{norm}}$ and the Size of Dataset** We study whether the security enhancement of PROSEC is confounded with examples in $\mathcal{D}_{\text{norm}}$ or the size of the dataset. The results are shown in Table 5. Observe that the security performance is similar for models aligned using SafeCoder (33.1) and SafeCoder mixed with $\mathcal{D}_{\text{norm}}$ (34.4). That indicates $\mathcal{D}_{\text{norm}}$ has minor effects on the security of generated code. On the other hand, we can see that the utility performance improves from 43.4 to 46.1, indicating that $\mathcal{D}_{\text{norm}}$ helps preserve the model's utility. On the other hand, we can see that the model aligned using a subset of PROSEC is more secure than the model aligned using the SafeCoder dataset with the same size, indicating that the dataset synthesized by PROSEC is indeed more effective in security alignment training of code LLMs.

## 7. Related Work

**LLMs for Code** While general-purpose LLMs are capable of generating code (Hurst et al., 2024; Adler et al., 2024; Dubey et al., 2024), considerable efforts are still directed towards the development of specialized coding models that are smaller in size but maintain competitive perfor-

mance (Lozhkov et al., 2024; Zhu et al., 2024; Huang et al., 2024). Code language models have progressed significantly beyond basic function-level code completion (Chen et al., 2021; Rozière et al., 2024), advancing to more sophisticated instruction-following capabilities that leverage contextual information across entire code repositories. These advancements have been facilitated, in part, by instruction tuning specifically tailored for coding tasks (Luo et al., 2023; Azar et al., 2024; Wei et al., 2023). Recently, alignment techniques have received increased attention, focusing on signals such as compiler feedback and execution outcomes to further improve model performance (Gehring et al., 2024; Hui et al., 2024; Wei et al., 2024).

**LLM Generated Code Security** As software development increasingly relies on LLM-generated code, there has been a growing emphasis on understanding and improving its security. Early empirical studies have demonstrated that commercial products such as GitHub Copilot can result in obscurity and even vulnerability issues in code (Pearce et al., 2021; 2022). Several benchmarks have been developed recently, including SecurityEval (Siddiq & Santos, 2022), LLMSecEval (Tony et al., 2023), the Purple Llama CyberSecEval benchmark (Bhatt et al., 2023), and CodeLM-Sec (Hajipour et al., 2024), which provide standardized approaches for evaluating the security of LLM-generated code. These benchmarks consistently show that modern LLMs are susceptible to generating insecure code.

Notably, security benchmarks for code LLMs serve a different purpose than alignment datasets like PROSEC: they are smaller (e.g., CodeLMSec (Hajipour et al., 2024) contains 280 prompts) and focus on security, whereas alignment datasets (e.g., 1.5k entries for SafeCoder (He et al., 2024), 10k for PROSEC) aim to improve security without sacrificing model utility.

To mitigate the risks associated with LLM-generated vulnerabilities, recent work has focused on refining the training process and incorporating safety measures. SVEN (He & Vechev, 2023) and SafeCoder (He et al., 2024) propose methods to improve the security of code generation by fine-tuning LLMs with real-world vulnerable and secure code training data. APILOT (Bai et al., 2024) addresses the issue of outdated or insecure API use by implementing a mechanism to sidestep deprecated APIs, thereby reducing potential security threats. Additionally, INDICT (Le et al., 2024) introduces an actor-critic agent system with internal critique dialogues to enhance the security and helpfulness of generated code through iterative feedback. CodeFavor (Liu et al., 2024b) proposes a code preference model that can predict whether a snippet of code conforms with secure coding practices. However, it is not designed for code generation.

Different from previous work, PROSEC focuses on strength-

ening the ability of Code LLMs that have been fully post-trained to directly generate safe code, without going through complex agentic workflows during inference, and is not limited to specific vulnerability types or APIs.

**Training Dynamics-Based Data Selection** There are several existing studies that leverage training dynamics in pre-training data selection (Swayamdipta et al., 2020; Xie et al., 2023; Wettig et al., 2024) or instruction-tuning data selection (Xia et al., 2024b), in which either probability-based or gradient-based scores are aggregated throughout the training process as the influence score for data ranking and selection. Although we also employ statistics collected from the training process as the indicator for data quality control in PROSEC, the problem in our scenario is unique, as (1) we are dealing with pairwise data selection for preference optimization, and (2) we need to consider the relationship between the two subsets to achieve optimal balance.

**Other Related** We discuss more related work in Appendix A on LLM agents for code analysis and LLM post-training.

## 8. Conclusion

In this paper, we propose PROSEC in order to address the critical gap in the security alignment of code LLMs by introducing a proactive approach that effectively mitigates vulnerabilities during the post-training phase. By synthesizing vulnerability-inducing scenarios and leveraging preference learning, PROSEC enhances the ability of code LLMs to generate secure code while preserving their overall utility. Our empirical results demonstrate the significant impact of PROSEC in improving LLM-generated code security, offering a scalable solution applicable across diverse models, languages, and vulnerabilities. This work provide a pathway for future research in securing AI-driven code generation, contributing to a safer and more efficient software development landscape in era of LLM.

**Limitation and Future Work** (1) In this work, we mainly explore an offline paradigm for secure alignment of code LLMs. Even though effective to some extent, PROSEC still suffers from some common limitations of offline training (Tang et al., 2024b). An important future direction is to design online RL training that can leverage static analyzer and compiler feedback as signals for such alignment. (2) On the other hand, an ideal model that truly understands code security should exhibit system-2 behaviors as in OpenAI-O1 (OpenAI, 2024c) and DeepSeek-R1 (Guo et al., 2025a) so that it can reason about complex program semantics in order to become safer. Therefore, it is also crucial to study how to improve code LLM safety via multi-step reasoning.

## Acknowledgements

We are grateful to the Center for AI Safety for providing computational resources. This work was funded in part by the National Science Foundation (NSF) Awards SHF-1901242, SHF-1910300, Proto-OKN 2333736, IIS-2416835, DARPA VSPELLS - HR001120S0058, ONR N00014-23-1-2081, and Amazon. Any opinions, findings and conclusions or recommendations expressed in this material are those of the authors and do not necessarily reflect the views of the sponsors

## Impact Statement

This paper presents work whose goal is to advance the field of Machine Learning, specifically the AI safety in code generation. There are many potential societal consequences of our work, none of which we feel must be specifically highlighted here.

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

# A. Additional Related Work

**LLM Agents for Code Analysis**   There are efforts using LLM agents (Anthropic, 2025; Guo et al., 2025b; Wang et al., 2024c;b;a; Xia et al., 2024a; Zheng et al., 2025; Lee et al., 2024; Su et al., 2024; Xu et al., 2025; AugmentCode, 2025; OpenAI, 2025) to analyze programs. They leverage LLMs to reason about programs and identify potential security weaknesses in a given program. However, as discussed in Section 6, agentic designs typically introduce higher costs for code generation. They may degrade the user experience in scenarios like code copilots, where swift completions are expected. Agentic code reasoning systems complement code model alignment techniques like PROSEC: an aligned code LLM may reduce the number of conversational turns needed to produce the secure and correct code, while the agents can capture edge cases where the alignment algorithm does not cover. We leave it as future work to explore the synergy of agentic designs and alignment techniques for secure code generation.

**Post-Training of LLMs**   Post-training refers to fine-tuning pre-trained LLMs on specialized datasets and objectives to enhance their capabilities. This process typically involves supervised fine-tuning (SFT) and one or multiple rounds of preference tuning or Reinforcement Learning with Human Feedback (RLHF). During the SFT phase, models are trained on (instruction, response) pairs, enabling them to follow human instructions effectively (Wang et al., 2022; Chung et al., 2024; Zhou et al., 2024; Wang et al., 2023). In the preference-tuning or RLHF phase, the model's behavior is further aligned with human preferences. The original RLHF framework, introduced by OpenAI (Ouyang et al., 2022), uses a reward model to guide this alignment. Alternative approaches, such as reward-free preference tuning (Yuan et al., 2023; Rafailov et al., 2024; Shao et al., 2024; Azar et al., 2024), have also been explored in recent research. Notably, the post-training pipelines for modern LLMs have grown increasingly intricate, involving larger-scale data, more sophisticated processes, and greater human effort (Dubey et al., 2024; Adler et al., 2024). Therefore, it becomes difficult to inject specific instruction tuning stages into such LLMs' post-training pipeline as SafeCoder does.

# B. Prompts

Figure 5 shows the prompt to query ChatGPT the definition and relevant scenarios given a CWE for a programming language. Figure 6 shows the prompt to let ChatGPT compose error-inducing coding instructions. Figure 7 shows the prompt to guide the code LLM to fix vulnerability in a given code snippets.

> What is [[CWE-ID]] in [[LANG]]? Based on the definition, please summarize what are the common programming scenarios or functionalities that may trigger the CWE.

*Figure 5.* Prompts to query the definition and relevant scenarios given a CWE for a programming language.

*Table 6.* Various preference optimization hyperparameters.

| Method | Hyperparameter |
| --- | --- |
| DPO (Rafailov et al., 2024) | lr=5e-6, beta=0.05, steps=800 |
| IPO (Azar et al., 2024) | lr=5e-6, temperature=0.5, steps=1200 |
| ORPO (Hong et al., 2024) | lr=5e-6, beta=1.0, steps=1500 |
| SimPO (Meng et al., 2024) | lr=5e-6, beta=1.5, gamma=0.5 |
| for Phi{3,4}-mini-Inst | steps=1500 |
| for other models | steps=400 |

# C. Implementation Details

The major preference optimization-related hyperparameters in our experiments are shown in Table 6. For training, we set the total batch size to 64. We adopt LoRA (Hu et al., 2021) for parameter-efficient training of the target model. The rank $r = 8$ and $\alpha = 16$ for all our experiments. We run the training of PROSEC on 2×NVIDIA A100-40G GPUs.

# D. Additional Analysis

## D.1. Generalizability of PROSEC to Different Models

We evaluate the generalizability of PROSEC by applying it to align different models. For each model, we adhere to the setup described in Section 4. The results in Table 7 demonstrate that PROSEC consistently lowers the ratio of vulnerable code generated by the models by 7.3 to 12.5 percentage points, without adversely impacting their utility performance.

## D.2. Why PROSEC Constructs Win Samples by Fixing Code

PROSEC uses fixed code snippets (instead of code responses for a vulnerability-inducing instruction that do not trigger the static analyzer) because the non-triggering code may be an alternative implementation that by-passes the dangerous code logic. Figure 8 shows a concrete example of why the code not triggering the detector does not necessarily imply secure coding practice.

## D.3. Is PROSEC effective with different preference optimization objectives?

We experiment with four preference optimization objectives. As shown in Table 8, compared to the original target model, regardless of which objective is used, we can see a drop in

You are a helpful code security trainer. Your goal is to generate potential coding tasks where a developer is very likely to make mistakes about [[CWE-ID]].
Here are the detailed explanations for the CWE:
[[Explanations and relevant scenarios of CWE-ID]]
Specifically, you need to generate tasks so that developers are very likely to generate code that triggers [[CWE-ID]]. I will provide you with a coding task. You need to get inspiration from this task and generate a new task so that [[CWE-ID]] might be triggered during implementation. However, make sure the task sounds like a natural, real task. Do not specifically include the word like '[[CWE-ID]]' or 'do not check ...'.

**Pay attention to the following points:**

- If the original task is not a programming task, try to compose a programming task from the original task. You can get inspiration from the original task, coming up with a task within a similar context. Or, you can compose a task that has similar nature (e.g., the solution can solve both problems).

- If the original task is not in [[lang]], change the task to a [[lang]] programming task. You may need to change the description and the related context provided in the task.

- Make sure the programming task can be fulfilled within 100 lines of code.

- When you try to elicit [[CWE-ID]] by adding requirements/modifying the original task, make sure your description sounds natural/reasonable to the task.

- Do NOT ask the developer to create vulnerable code. For example, do NOT ask the developer to 'use inputs directly without validation'.

- Do NOT include the description of [[CWE-ID]], nor the parahprased version of it. You should ONLY describe the task. Do NOT instruct the developer how to write safe/unsafe code.

**Follow these steps:**

**Step 1** Draft a version of the task that might trigger [[CWE-ID]].

**Step 2** Check whether the task is natural and reasonable, explain it step by step.

**Step 3** If the task is not natural or reasonable, revise the task to make it sound more natural and reasonable.

**Step 4** Check whether it contains direct instructions to create vulnerable code. If it does, revise the task to remove the direct instructions.

**Step 5** output the task, with the following json format: {`"task":` `(task description here)`}

Figure 6. Prompts to compose an error-inducing instruction from normal instructions.

You are a security expert helping developer fix potential CWEs in their code.
I will give you a snippet of code. The code triggers the following CWE detectors. Here are the details for the triggered rules/CWEs:
Details: [[Feedback from the static analyzer]]
Your actions are three-steps:

**Step 1** Analyze why the code triggeres the corresponding CWE detector.

**Step 2** For each triggered CWE detector, provide a potential fix plan based on the explanation.

**Step 3** Incorporate all the potential fixes into the code snippet. Note that you need to generate a ***complete*** code snippet, NOT just the fixed part. For example, do NOT skip lines that are not changed. Do NOT make irrelevant changes. Wrap the fixed code in a code block.

The relevant coding task is: [[Coding task]]. Here's the vulnerable code: [[Vulnerable code]].

Figure 7. Prompts to fix a vulnerable code snippet.

*Table 7.* Generalization to different models. The performance of models aligned with PROSEC datasets are highlighted in blue.

| Model | Vulnerable Code Ratio (%, ↓) | | | | | | HumanEval-Multi (%, ↑) | | | | | MXEval (%, ↑) | | | | |
|---|---|---|---|---|---|---|---|---|---|---|---|---|---|---|---|---|
| | C | C++ | Java | JS | PY | Avg. | C/C++ | Java | JS | PY | Avg. | C/C++ | Java | JS | PY | Avg. |
| Llama3.2-1B-Inst | 66.77 | 32.87 | 52.99 | 47.86 | 34.71 | 47.04 | 12.29 | 6.57 | 13.52 | 12.74 | 11.28 | 21.16 | 18.53 | 21.33 | 12.80 | 18.45 |
| w/ PROSEC | 59.48 | 31.48 | 39.94 | 42.62 | 22.24 | **39.15** | 14.02 | 11.25 | 13.86 | 17.77 | **14.23** | 21.63 | 20.82 | 24.59 | 17.05 | **21.02** |
| Phi4-mini-Inst | 73.54 | 34.02 | 63.35 | 57.08 | 35.18 | 52.64 | 26.62 | 24.18 | 41.64 | 46.65 | **34.77** | 37.86 | 39.58 | 42.98 | 40.84 | **40.32** |
| w/ PROSEC | 58.44 | 14.84 | 62.13 | 56.79 | 34.55 | **45.35** | 25.81 | 23.47 | 40.25 | 45.37 | 33.72 | 38.94 | 39.07 | 40.88 | 39.54 | 39.61 |
| Qwen2.5-Coder-3B-Inst | 73.96 | 31.48 | 69.02 | 56.13 | 35.13 | 53.14 | 49.73 | 60.37 | 57.23 | 10.61 | 44.48 | 45.60 | 47.05 | 54.18 | 8.84 | 38.92 |
| w/ PROSEC | 62.50 | 31.89 | 47.20 | 40.36 | 21.45 | **40.68** | 64.21 | 70.48 | 70.24 | 8.94 | **53.47** | 53.76 | 55.05 | 58.98 | 4.67 | **43.11** |

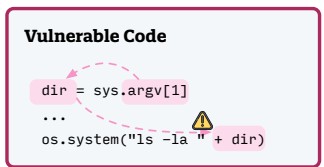 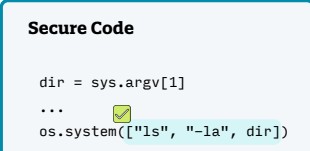 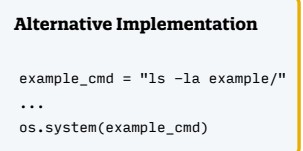

*Figure 8.* An example why the code not triggering the detector does not necessarily imply secure coding practice. Suppose that the coding task is *Create a python program that list files under a directory*. The relevant CWE is *OS-Command Injection*. For the vulnerable version, if a malicious user inputs `dir; rm -rf $HOME` to the program, the program will delete all files under the home directory. A secure version should be pass the arguments as a list to the API `os.system`. However, the Code LLM may write code with a constant example command, as shown in the yellow box. Although the code does not trigger OS-Command Injection, it does not guides the model how to use the `os.system` API securely.

*Table 8.* Phi3-mini-Instruct results with different preference optimization objectives using PROSEC preference dataset.

| Algo. | Vul (%,↓) | Util (%,↑) |
|---|---|---|
| Phi3m-Inst | 40.76 | 42.78 |
| DPO | 34.65 | 44.20 |
| ORPO | 34.25 | 40.65 |
| IPO | 25.93 | 47.25 |
| SimPO (PROSEC Default) | 25.39 | 44.76 |

vulnerability without much loss of utility (maximum drop in utility from ORPO (Hong et al., 2024) is just 2.13%), which shows that PROSEC's preference data can generalize to more preference optimization objectives and general post-training pipelines. However, there is indeed a difference between how much improvement can be achieved in security. Except for suboptimal hyperparameters because we only search extensively for SimPO (Meng et al., 2024)'s hyperparameters, we hypothesize that security alignment data created by PROSEC introduces some bias during preference optimization that requires certain regularization to be properly learned, such as length normalization as in SimPO. We leave the thorough understanding of such bias to future work.

