# OpenReview forum: "ProSec: Fortifying Code LLMs with Proactive Security Alignment"
_ICML.cc/2025/Conference — ICML 2025 poster_

### Official Review · Reviewer_VaQK · 2025-03-09

**Overall Recommendation:** 3

**Summary:**

The paper introduces PROSEC, a method for proactively identifying weaknesses in code-generating AI models by creating specific coding scenarios that are likely to introduce vulnerabilities. PROSEC creates a significantly larger dataset of vulnerability-inducing situations compared to previous methods. Experiments compare different code models with PROSEC and test their ability to perform regular coding tasks.

## update after rebuttal
Thank you again for the detailed explanations in the rebuttal regarding the difference from related work.

**Claims And Evidence:**

I could not identify specific unclear claims, but check the comments section for more details.

**Essential References Not Discussed:**

Many references are generally published in that domain, but no obvious paper is missing, as far as I can tell.

**Ethical Review Concerns:**

Not necessarily critical, but since the paper is about security, I would expect some comment on that in the paper.

**Experimental Designs Or Analyses:**

The experimental design is described and contains details necessary to understand the results. However, aspects of the verification such as the utility and other details remain unclear.

**Methods And Evaluation Criteria:**

The proposed method seems reasonable from a high level. However, the concept is not new, so the contribution is not really novel.

**Other Comments Or Suggestions:**

Thank you for submitting the paper. It elaborates on a very timely topic. Therefore, research in this domain remains required to build more secure code-generative LLMs.

**The idea**:

Unfortunately, the novelty of the paper is unclear. This paper [new1] uses a similar approach for the same purpose. Therefore, I would expect a comparison with this method and a detailed elaboration on the differences to this method for benchmarking code LLMs.

In Section 2, the paper claims “…to reduce the likelihood of generated code being detected by the static analyzer…”. This sounds like the paper’s target is to prevent detection. However, a better ultimate goal would be to prevent the generation of vulnerable code or at least to fix the code. Does the paper really try to prevent detection?

**The presentation:**

What is preferred/win and a less preferred/lose? This is mentioned in Section 2 after Equation 2.

Figure 1 misses some more detailed explanations. It may become clearer when reading the paper. However, I am missing a designated section for explaining the details of the figure.

The paper measures the functionality of the generated code. However, there are no details on how the functionality is assessed. This is generally not an easy task and several different strategies are possible. What is used in this paper?

It remains unclear why data needs to be selected for the evaluation in Section 3.3.

It also remains unclear why an optimizer is required. Further, at the beginning of Section 5, it is not explained how the results are reported, specifically how and why at least 20% remain in the dataset.

Effects on model utility: Where is this shown? The paper does not point to a result table or plot.

Also, it is not mentioned which and how many CWEs are considered and how the selection is justified.

Further, I am missing an explanation of how the diversity of the code is measured and assessed.

[new1] Hajipour et al. “CodeLMSec Benchmark: Systematically Evaluating and Finding Security Vulnerabilities in Black-Box Code Language Models,” SatML 2024

**Other Strengths And Weaknesses:**

Strengths:

- Timely topic

Weaknesses:

- The presentation of the paper needs improvement
- The novelty remains unclear

**Questions For Authors:**

- Does the paper really try to prevent detection?
- What is preferred/win and a less preferred/lose?
- What strategy to meassure the functionality of code is used in this paper?
- Why is there a selection of data in Section 3.3?
- Effects on model utility: Where is this shown?

**Relation To Broader Scientific Literature:**

The paper has a good overview of related work. However, critical related work is missing.

**Theoretical Claims:**

N/A

---

> ### Author Rebuttal · Authors · 2025-03-30
>
> We appreciate your feedback and respectfully clarify key differences between CodeLMSec and ProSec to illustrate our unique contributions:
>
> ## Different goals
> CodeLMSec and ProSec serve fundamentally different purposes. CodeLMSec is a codeLM security **benchmark** that evaluates codeLMs with vulnerability-inducing prompts. In contrast, ProSec is an **alignment training technique** that employs scalable data synthesis to produce a high-quality dataset, effectively securing codeLMs without harming utility.
>
> Each entry in the ProSec dataset includes a vulnerability-inducing prompt, an insecure implementation, and a secure implementation. The alignment training loss requires a model to generate secure implementation with a higher probability than generating the insecure one.
>
> ## Scalability
> Effective alignment requires more samples than a typical benchmark. While CodeLMSec includes 280 prompts, the previous SOTA alignment dataset contained 1.5k prompts. ProSec scales up to more than 10k entries. ProSec automates the data synthesis process by composing diverse vul-inducing scenarios using CWE definitions and existing instruction-tuning datasets, avoiding the labor-intensive manual curation required by CodeLMSec.
>
> ## Reference code
> Benchmarks need only prompts; alignment requires paired insecure and secure code. ProSec uses rejection sampling with static analyzers to curate insecure code snippets, and it instructs LLMs to produce secure fixes, thereby constructing pairwise alignment data entries.
>
> ## Balancing security and utility
> An alignment dataset needs to balance between enhancing security and preserving utility. ProSec proposes a training dynamic-based data selection algorithm to construct a utility-preserving dataset, ensuring models’ utility is not compromised during security alignment.
>
> Technically, we follow standard practices by evaluating model utility on coding benchmarks (i.e., HumanEval and MXEval) and assessing correctness via test cases. E.g., Table 1 shows that for Phi3mini-Inst, ProSec achieves 44% on MXEval versus SafeCoder's 42%, showing better utility preservation.
>
> ## Empirical comparison
> Although the two works are different, we did experiments during the rebuttal to show unique challenges in synthesizing alignment data.
>
> We used the prompts from CodeLMSec to construct an alignment training dataset with the same pipeline as ProSec. (The coding instructions were obtained from CodeLMSec rather than synthesized by ProSec.) The ratios of vulnerable code generated are (lower is better): the original model: 45.6, ProSec: 19.7, CodeLMSec: 38.9. We can see that the model aligned with ProSec is more secure, underscoring the challenge of effective alignment data synthesis.
>
> ## Clarifications
>
> > Q1: What is preferred/win and a less preferred/lose?
>
> Each entry in an alignment dataset consists of three parts: a prompt, a preferred (or “win”) response, and a less-preferred (or “lose”) response [1]. In ProSec, the prompt is a vulnerability-inducing prompt, while the “win” and “lose” responses correspond to secure and insecure implementations, respectively.
>
> > Q2: The goal of ProSec?
>
> The goal of ProSec is to prevent the generation of vulnerable code. It trains the model to favor generating the secure code over the insecure ones. We will revise section 2.
>
> > Q3: How to measure functionality?
>
> We follow established practices[2, 3, 4] to set up the experiments. The functionality correctness is evaluated by test cases. Code generations that pass all test cases are considered correct.
>
> > Q4: Why is data selection necessary?
>
> The data selection is critical for balancing security and utility of the aligned model. Intuitively, including too many utility-preserving data samples weakens the security alignment, while too few impairs the model’s utility. ProSec employs a training dynamic-based algorithm to identify and selectively include those utility-preserving data samples whose distribution is disrupted by the security alignment.
>
> > Q5: How to measure utility?
>
> As shown in Table 1, the utility is measured as the performance on two coding benchmarks, HumanEval and MXEval.
>
> > Q6: What and how are CWEs selected?
>
> We select 38 CWEs that overlap between PurpleLlama and SafeCoder to set up a fair evaluation. Please see Section 4 (line 269) for details.
>
> > Q7: How to measure data diversity?
>
> We use the cosine similarity of semantics embeddings to analyze the diversity of the dataset (line 359, Fig. 4). We use string editing distance (line 255, Section 3.3) to deduplicate code snippets and thus increase dataset diversity.
>
> [1] ​​Rafailov, Rafael, et al. Direct preference optimization: Your language model is secretly a reward model. NeurIPS’23
>
> [2] He, Jingxuan, et al. Instruction tuning for secure code generation. ICML’24.
>
> [3] He, Jingxuan,et al. Large language models for code: Security hardening and adversarial testing. CCS’23
>
> [4] Wei, Yuxiang, et al. Magicoder: Empowering code generation with oss-instruct. ICML’24.

---

> > ### Comment · Reviewer_VaQK · 2025-04-02
> >
> > Thank you for the detailed rebuttal and for answering my questions. Since I am the negative reviewer here, I want to emphasize (which I missed before) that my main concern is mainly the novelty and the difference from related work. However, I feel that this could be addressed in the rebuttal. Specifically, the goal is to **prevent** the generation of vulnerable code. I would like to ask you to clarify a few things in the paper to make it more accessible to the reader. Having that said, I am happy to increase my score.

---

> > > ### Author Response · Authors · 2025-04-04
> > >
> > > Thank you for your insightful feedback. We will update the paper to include a more detailed discussion on the differences between ProSec and related work, emphasizing its unique contributions. We appreciate your constructive suggestions and your willingness to increase your score.

---

### Official Review · Reviewer_gVNW · 2025-03-13

**Overall Recommendation:** 3

**Summary:**

This work enhances the security of code LLMs by proposing the PROSEC framework. PROSEC is an automated pipeline designed to synthesize code security-related preference data. It consists of three stages: 1) Construct instructions that induce insecure code based on Common Weakness Enumerations (CWEs)  and ensure diversity of the instructions through a clustering method. 2) Analyze the responses for insecure content using a static analyzer. If insecure content is detected, it is regarded as a negative sample. Subsequently, the LLM is prompted to modify the response to remove the insecure content, and the modified response is regarded as a positive sample. 3) To ensure LLMs' utility, this work introduces normal preference data and retains the most impactful data. Experimental results show that the preference pairs constructed using the PROSEC framework, when applied with the SimPO alignment algorithm, can enhance the security of the code LLMs while maintaining their utility.

**Claims And Evidence:**

Yes, the claims made in the submission are supported by clear and convincing evidence.

**Essential References Not Discussed:**

No

**Experimental Designs Or Analyses:**

Yes, Ic check the soundness of experimental designs.

**Methods And Evaluation Criteria:**

Yes, proposed methods make sense for the problem at hand.

**Other Comments Or Suggestions:**

None

**Other Strengths And Weaknesses:**

Strengths:
1) The motivation is clear. The study designs an automatic pipeline for synthesizing code preference data and considers the diversity of instructions and the issue of data quality during the design.
2) The results are good. We note that the method proposed in this study significantly improves the security of the code LLMs while ensuring their usability.

Weakness:
1) Although the method proposed in this work is effective, it is essentially a simple data synthesis method. This feels more like an engineering-focused work, and I am somewhat concerned whether this can be published at ICML.
2) The focus of this work should be on synthesizing higher-quality code preference data, but I am concerned about whether the introduction of normal preference data has had an impact. You should integrate the normal preference data into Safecoder or remove it from your method to conduct an ablation study.

**Questions For Authors:**

None

**Relation To Broader Scientific Literature:**

The key contributions of the paper are good.

**Theoretical Claims:**

Yes, I check the correctness of any proofs for theoretical claims.

---

> ### Author Rebuttal · Authors · 2025-03-30
>
> Thank you for your detailed and supportive review.
>
> ## ProSec’s relevance to ICML
>
> We respectfully contend that our work aligns well with previous contributions recognized at ICML. For example, prior works such as data selection for language model training [Qurating, ICML’24], data synthesis for code language models [MagicCoder, ICML’24] and security-focused SFT for code language models [SafeCoder, ICML’24] introduced innovative practices in LLM training and alignment, thereby establishing a strong precedent for valuable engineering-focused work.
>
> Furthermore, we note that ICML has a tradition of supporting application-driven research. For instance, [Invariant; ICML’23] prompts LLMs to generate loop invariants – addressing a significant challenge in the software engineering domain; [Next; ICML’24] enhances program repair performance by incorporating execution trace information to prompts; and [StackSight; ICML’24] leverages LLMs to translate low-level code into readable C++ code via CoT prompts.
>
> Similarly, ProSec offers a scalable approach to securing code language models during the alignment stage. In particular, our introduction of utility-preserving dataset and our formulation of alignment side effects via training dynamics provide valuable technical insights into balancing security alignment and model utility. We believe these contributions address important practical challenges and pave the way for further advancements in aligning codeLM with domain specific constraints.
>
> ## Ablation study for normal preference data
>
> We appreciate the suggestions and will include the following discussion in the paper.
>
> During rebuttal, we conducted experiments to illustrate the effectiveness of introducing the utility-preserving dataset (DNorm).
> Due to time constraints, we evaluate all models on a subset of PurpleLlama and MXEval for security and utility assessment, respectively. The results are summarized below:
>
> | Model              | Vul(%) | Util(%) |
> |--------------------|--------|---------|
> | The original model | 40.8   | 42.8    |
> | SafeCoder          | 33.1   | 43.4    |
> | SafeCoder+DNorm    | 34.4   | 46.1    |
> | ProSec         | 25.0   | 45.2    |
> | ProSec w/o DNorm   | 4.1    | 3.1     |
>
> We can see that DNorm indeed helps preserve the models’ utility. By adding DNorm to the SafeCoder dataset, the utility of the aligned model improves slightly while maintaining a comparable level of security performance compared to the vanilla SafeCoder dataset. In contrast, when DNorm is removed from the ProSec dataset, the model’s utility performance drops dramatically, indicating that aligning the model solely on the security dataset would significantly compromise its utility.
>
>
> [Qurating] Wettig, Alexander, et al. "Qurating: Selecting high-quality data for training language models." ICML’24
>
> [MagicCoder] Wei, Yuxiang, et al. Magicoder: Empowering code generation with oss-instruct. ICML’24.
>
> [SafeCoder] He, Jingxuan, et al. Instruction tuning for secure code generation. ICML’24.
>
> [Invariant] Pei, Kexin, et al. "Can large language models reason about program invariants?."ICML’23
>
> [Next] Ni, Ansong, et al. "Next: Teaching large language models to reason about code execution." ICML’24
>
> [StackSight] Fang, Weike, et al. "StackSight: Unveiling webassembly through large language models and neurosymbolic chain-of-thought decompilation." ICML’24

---

> > ### Comment · Reviewer_gVNW · 2025-04-03
> >
> > Thank you for your comprehensive response. I would like to clarify that I am more interested in seeing the improvements in code security achieved **solely by using Dnorm**. In simple terms, I want to understand how much of the performance improvement comes from Dnorm and how much comes from the synthetic data you constructed. However, based on the results you presented, it seems that **Dnorm has played a significant role in enhancing security**. Does this mean that the significance of the data you constructed is diminished? In other words, could we improve model security by constructing more normal data instead?

---

> > > ### Author Response · Authors · 2025-04-04
> > >
> > > Thank you for the detailed question. **We would like to clarify that both security-focused data (DSec) and utility-preserving data (DNorm) are synthesized by the ProSec pipeline. Simply adding more normal (DNorm) data does not improve model security.** Intuitively, an excessive number of utility-preserving data samples (DNorm) can dilute the security alignment, while too few can impair the model’s utility. ProSec employs a training dynamic-based algorithm to identify and selectively include those utility-preserving data samples whose distribution is disrupted by the security alignment.
> > >
> > > The key technical contributions of ProSec are (1) synthesizing DSec that **enhances the model’s security performance** with diverse security-focused coding scenarios, (2) synthesizing DNorm that **preserves the model’s utility** during the security alignment process, and (3) proposing a data selection algorithm that achieves better balance between the security enhancement and utility preservation.
> > >
> > > We appreciate the question and will add the following discussion to the paper to clarify this further.
> > >
> > > Following are the details.
> > >
> > > ## Clarification on Metrics
> > >
> > > Following established practices[1,2], we evaluate a security alignment dataset from two perspectives: (1) Security performance, measured as the ratio of vulnerable code generated. **A lower percentage indicates better security.** (2) Utility performance, measured by the model’s performance on coding tasks, with a higher score indicating better utility preservation.
> > >
> > > ## Clarification on Results
> > >
> > > **Our results indicate that DNorm primarily supports utility preservation rather than enhancing security.**
> > >
> > > For example, consider the rows “SafeCoder” and “SafeCoder + DNorm” in the table we shared during the initial rebuttal. The model aligned without DNorm achieved a security performance of 33.1, compared to 34.4 when DNorm was included. Although the security metrics remain relatively similar, the inclusion of DNorm clearly benefits utility performance, with scores rising from 43.4 to 46.1.
> > >
> > > A similar trend is observed with the ProSec dataset (rows “ProSec” and “ProSec **w/o** DNorm”). In this instance, the model aligned without DNorm achieved a better security performance (4.1) than its counterpart with DNorm (25.0). However, the utility dropped substantially – from 45.2 to 3.1 – when DNorm was omitted. The results show that DNorm’s role is not to enhance security but to mitigate the adverse effects on utility that can arise during security alignment.
> > >
> > > Further evidence is provided by the trends shown in Table 2 (line 385) of the paper. The key results are as follows:
> > >
> > > | Configuration     | Vul(%) | Util(%) |
> > > |-------------------|--------|---------|
> > > | DSec+10%DNorm     | **5.9**    | 15.3    |
> > > | DSec+30%DNorm     | 27.5   | 42.1    |
> > > | DSec+70%DNorm     | 25.6   | **45.1**    |
> > >
> > > As the proportion of DNorm increases—from 10% to 70%—utility performance improves (from 15.3 to 45.1), while security performance shifts (from 5.9 to 25.6). The results demonstrate that simply increasing the amount of DNorm data does not lead to enhanced security; rather, it primarily preserves the model's utility while striking a balance between security and performance. Note that the security metric exhibits a minor (<2%) increase when DNorm is raised from 30% to 70%. This slight variation is likely due to randomness in the sampling of DNorm subsets, given that the change is much smaller compared to the shift observed between 5.9 and 27.5.
> > >
> > > We hope the above discussion clarifies potential misunderstandings. We will add the discussion to our paper.
> > >
> > > [1] He, Jingxuan, et al. Instruction tuning for secure code generation. ICML’24.
> > >
> > > [2] He, Jingxuan,et al. Large language models for code: Security hardening and adversarial testing. CCS’23

---

### Official Review · Reviewer_CfLT · 2025-03-15

**Overall Recommendation:** 4

**Summary:**

The paper proposes ProSec (Proactive Security Alignment), an approach to align code LLMs with secure coding practices.
* It exposes the vulnerabilities by synthesizing error-inducing scenarios from Common Weakness Enumerations (CWEs) and generates fixes to vulnerable code.
* Models are then trained with preference learning objectives

The proposed synthesis procedure trigger more vulnerable code and resulting in a much larger dataset than previous work. Models trained with ProSec are significantly more secure without degrading performance.

### update after rebuttal ###
The authors have addressed my concerns and questions. I'm keeping my score as "accept".

**Claims And Evidence:**

The claims are well-supported.

**Essential References Not Discussed:**

N/A

**Experimental Designs Or Analyses:**

Yes, the experimental designs and analyses are valid and thorough.

**Methods And Evaluation Criteria:**

Yes, the evaluations for the proposed ProSec framework make sense.

**Other Comments Or Suggestions:**

N/A

**Other Strengths And Weaknesses:**

Strengths:
* The proposed method can greatly enrich datasets of vulnerable code.
* It shows that with (offline) preference optimization, models post-trained on the ProSec generated dataset outperforms a baseline dataset in security, without degrading capabilities.

Weaknesses:
* the vulnerabilities are limited to the CWE (common weakness enumerations) set.

**Questions For Authors:**

Is the improvement only affected by the fact that the generated dataset is 7x larger than SafeCoder? Can you do a control experiment where both datasets have the same size (and same secure / vulnerable mixture ratio), and test if the data quality of ProSec is also higher (e.g. more diverse)?

**Relation To Broader Scientific Literature:**

The main contribution lies in the method for constructing a relatively diverse vulnerable code dataset, across many types of weaknesses, tasks and coding languages.

**Theoretical Claims:**

N/A

---

> ### Author Rebuttal · Authors · 2025-03-30
>
> Thank you for the supportive review.
>
>
> >Is the improvement only affected by the fact that the generated dataset is 7x larger than SafeCoder? Can you do a control experiment where both datasets have the same size (and same secure / vulnerable mixture ratio), and test if the data quality of ProSec is also higher (e.g. more diverse)?
>
> The quality of ProSec dataset is higher than SafeCoder because ProSec scales to more diverse scenarios without requiring manual efforts.
>
> SafeCoder constructs alignment data entries by collecting real-world vulnerabilities and the corresponding fixes. However, real-world vulnerabilities and their fixes are sparse. For example, SafeCoder only collects 465 entries from 145 million git commits.
>
> On the other hand, ProSec leverages LLMs to enumerate potentially vulnerable-inducing scenarios, systematically scaling up to more diverse scenarios. Therefore, the overall quality of the ProSec dataset is better than that of the SafeCoder dataset.
>
> ## Empirical Support
>
> During rebuttal, we did another set of experiments as suggested by the reviewer.
> - **[Size]** We randomly sample a subset of the ProSec dataset to match the size of the SafeCoder dataset and run the alignment algorithm again. The resulting model is noted as “ProSec[Subset]”.
> - **[Mixture ratio]** The original SafeCoder dataset does not contain utility-preserving data for preference optimization. To make sure the ratio of security-focused and utility-preserving data samples are the same between ProSec and SafeCoder, we mix the vanilla SafeCoder dataset with the utility-preserving dataset of ProSec at the same ratio. The resulting model is noted as “SafeCoder+DNorm”.
>
> The empirical results show that the ProSec dataset is better than the SafeCoder dataset when both datasets contain the same number of entries and when both datasets contain the same ratio of security/utility data samples.
>
> Note that we evaluate all models on subsets of PurpleLlama and MXEval for security and utility measurements due to time constraints.
>
> | Model               | Vul(%) | Util(%) |
> |:---------------------|--------|---------|
> | The original model  | 40.8   | 42.8    |
> | SafeCoder           | 33.1   | 43.4    |
> | SafeCoder+DNorm     | 34.4   | 46.1    |
> | ProSec[Subset]      | 28.9   | 47.0    |
> | ProSec[Full]        | 25.0   | 45.2    |
>
> We can see that the model trained on the ProSec subset is safer than that trained on the SafeCoder dataset with the same size. It also outperforms the SafeCoder dataset mixed with the same ratio of benign data samples. That indicates the data quality of ProSec is indeed higher.
> Moreover, the model trained on the full ProSec dataset is safer than that trained on the subset, as expected. That is because the full ProSec dataset covers more scenarios.
>
> The alignment training on all datasets do not significantly affect coding utilities.
>
> We will include the experiments in the paper.

---

> > ### Comment · Reviewer_CfLT · 2025-04-04
> >
> > Thank you for your response and the additional experiment! It has addressed my concerns, and it'll be great to include this experiment in the paper. I'll keep my score, which is already an accept.

---

### Official Review · Reviewer_Fjvi · 2025-03-24

**Overall Recommendation:** 3

**Summary:**

This work proposes ProSec, a LLM-based framework to generate synthetic preference/alignment data containing security vulnerabilities using CWEs (Common Weakness Enumerations) data. Authors demonstrate that models (Phi3-mini-Inst and CodeLlama-7B-Inst) trained (SimPO) with ProSec alignment data produce code that is more secure in comparison to training via state of the art safecoder dataset. Ablation studies demonstrate preservation of utility.

Broadly the idea of aligning LLMs for security is novel and the approach to generating synthetic preference data using CWEs is interesting, although the technique used by prompting chatGPT (line 283 mentions claude-3.5-haiku, not clear if this is used).

The dependence on GPT like model is heavy. Authors use chatGPT to generate a vulnerability inducing instruction using the CWEs, use the instruction to generate code that has a security vulnerability, and also use the model to correct the vulnerability.

**Claims And Evidence:**

Models aligned with ProSec data more secure than safecoder dataset on vulnerable code ratio scores
- Phi3-mini-Inst models (28.86% vs. 44.72% )
- CodeLlama-7B-Inst models (28.55% vs. 40.33%)

Utility preservation: ProSec achieves security improvements with minimal impact on model performance on regular benchmarks.
Differences in performance on coding benchmarks < 2%.
Good performance on multi-lingual HumanEval and MBPP benchmarks.
Coverage across multiple programming languages (C/C++, Java, JavaScript, Python)

**Essential References Not Discussed:**

NIL

**Experimental Designs Or Analyses:**

Yes

**Methods And Evaluation Criteria:**

Yes

**Other Comments Or Suggestions:**

None

**Other Strengths And Weaknesses:**

Strengths
- novel idea to align LLMs for security by proactively generating synthetic data using CWEs (vs relying on previous approaches that construct datasets of vulnerable code and corresponding fixes from GitHub commits which can be quite sparse - 465 samples from 145 million git commits)

Weakness
- heavy reliance of ChatGPT like models for synthetic data generation. No open-source models in conjunction with agentic approaches were attempted.

**Questions For Authors:**

- Could you please explain what static analysers are used to validate generated (a) code with security vulnerabilities (b) fixed code.

- Could you please explain how the alignment data is filtered and what type of samples are removed in this process.

- Is the assumption that all CWEs are detectable via static analyzers. If so, why can't one continue to use existing LLMs (without alignment) and simply run the static checks post generation to check if code has security vulnerabilities. How does this work compare with other possible approach that might run the static checkers post generation via regular LLM and iteratively fix vulnerabilities using ChatGPT.

- what was the cost of generating synthetic data by querying ChatGPT.

**Relation To Broader Scientific Literature:**

I have not seen similar work on alignment for security.

**Theoretical Claims:**

NIL

---

> ### Author Rebuttal · Authors · 2025-03-30
>
> Thank you for the supportive and detailed review. We will include the discussions below to the paper.
>
> ## Q1: Explain static analyzers
>
> We use the static analyzers in PurpleLlama. It consists of three tools: regular expressions, semgrep, and weggli.
> All tools work on the generated source code. Regular expressions simply match insecure patterns in code (e.g., matching deprecated APIs). Semgrep and weggli first build the AST from a code snippet, and then search for problematic patterns (e.g., a potential NULL pointer that is dereferenced without being checked).
>
> ## Q2: How alignment data is filtered
> The alignment dataset contains two parts: the secure practice preference data (DSec) and the utility preservation preference data (DNorm). For both dataset, a data entry contains a prompt, a preferred code snippet, and a less-preferred code snippet.
>
> We control the quality of DSec by heuristics. We first check the syntactic correctness of the code snippets. Then we use static analyzers to make sure the preferred code is secure. After that, we use heuristics to make sure the preferred (secure) code snippet has corresponding functionality with the vulnerable code. Finally, we use string similarity (based on editing distance) to deduplicate data entries.
>
> For DNorm, we develop a data selection algorithm that takes training dynamics into consideration. Intuitively, the algorithm identifies and prioritizes the data entries whose utility are broken during the security alignment to mitigate degradation in utility. Specifically, we first align the target model with DSec only. For each candidate entry in DNorm, the algorithm keeps track of how its generation probabilities change during the alignment training. Then the algorithm selectively includes the entries whose generation probabilities decrease significantly during the security training.
>
> Please refer to section 3.3 (line 240) in the paper for details.
>
> ## Q3: Comparison with post-processing
> Yes, we focus on CWEs that can be detected by static analyzers. There are more complex CWEs that rely on global information or knowledge about program functionality. Detecting such CWEs are open challenges[1,2]. We leave it as future work to improve alignments on complex CWEs.
>
> An agentic workflow incurs higher computational costs and increased latency because it runs a static analyzer for every coding request and may require multiple queries to the code language model.
>
> This design could degrade the user experience in scenarios like code copilots, where swift completions are expected.
>
> In fact, post-processing with an agentic design complements code model alignment techniques: an aligned codeLM may reduce the number of conversational turns needed, while the agent can capture edge cases where the model produces insecure code.
>
> We further use empirical results to show the cost an agentic workflow may introduce.
> During rebuttal, we additionally implemented an agentic baseline that uses static analyzers to check generated code and iteratively asks the code generation model to fix. For each fix request, we provide the feedback from the static analyzers, the problematic code, and the initial coding instruction to the model. We evaluate the agentic workflow on a randomly sampled subset of PurpeLlama due to time constraints. Here are the statistics:
>
> | Max Fix Attempts | Success Rate (%) |
> |------------------|------------------|
> | 3                | 68.6             |
> | 5                | 73.7             |
> | 10               | 80.6             |
>
> Moreover, on average, a coding request requires five rounds of fixes to achieve the secure performance of ProSec. This demonstrates that using a security-aligned model is more efficient than simply applying post-processing to a secure code generation agentic workflow.
>
> Note that we choose to use the tested codeLM to fix the code, instead of using black box LLMs. That is because in a realistic use scenario of a smaller code LM, querying a larger black box model might be less preferred (due to the latency and cost) or even not allowed (due to privacy and policy concerns).
>
> ## Q4: Cost
>
> We use Claude-3.5-haiku to synthesize the instructions. For each CWE, we synthesize ~10k initial instructions, and then cluster them to identify the most diverse 2k instructions. The cost to synthesize instructions for each CWE is around 5 USD.
>
> [1] Ding, Yangruibo, et al. "Vulnerability detection with code language models: How far are we?." arXiv preprint arXiv:2403.18624 (2024).
>
> [2] Google, Project Zero. https://googleprojectzero.blogspot.com/2024/06/project-naptime.html

---

### Decision · Program_Chairs · 2025-05-01

**Decision:**

Accept (poster)

**Comment:**

The reviewers agreed that the data synthesis approach in this work is reasonably novel and produces a valuable dataset. The methodology is well-designed and the paper includes sufficient evaluations & ablations. Overall, this is a valuable contribution to improving the security of LLM-generated code.